# Universal salt-assisted assembly of MXene from suspension on polymer substrates

Liang Zhao [1,6], Lingyi Bi [2,6], Jiayue Hu [3,6], Guanhui Gao[4], Danzhen Zhang [2], Yun Li[1], Aidan Flynn[1], Teng Zhang[2], Ruocun Wang [2], Xuemei M. Cheng [5], Ling Liu [3] ✉, Yury Gogotsi [2] ✉ & Bo Li [1] ✉

Two-dimensional carbides and nitrides, known as MXenes, are promising for water-processable coatings due to their excellent electrical, thermal, and optical properties. However, depositing hydrophilic MXene nanosheets onto inert or hydrophobic polymer surfaces requires plasma treatment or chemical modification. This study demonstrates a universal salt-assisted assembly method that produces ultra-thin, uniform MXene coatings with exceptional mechanical stability and washability on various polymers, including high-performance polymers for extreme temperatures. The salt in the $Ti_3C_2T_x$ colloidal suspension reduces surface charges, enabling electrostatically hydrophobized MXene deposition on polymers. A library of salts was used to optimize assembly kinetics and coating morphology. A 170 nm MXene coating can reduce radiation temperature by ~200 °C on a 300 °C PEEK substrate, while the coating on Kevlar fabric provides comfort in extreme conditions, including outer space and polar regions.

MXenes have emerged as a large family of conductive two-dimensional (2D) materials in the past decade[1]. Recent findings have shown that MXene nanosheet films have outstanding thermal properties, including a wide range of emissivity in the mid-infrared (IR) spectrum, from very low to very high, along with low thermal conductivity in the out-of-plane direction[2]. As a result, MXene nanosheet films can provide thermal shielding or insulation at submicrometer thickness, having weight orders of magnitude smaller than conventional insulating materials, and work at elevated temperatures. With the IR emissivity at the level of polished metal and at least two orders of magnitude lower thermal conductivity, $Ti_3C_2T_x$ nanosheet film offers exceptional infrared radiation screening capability[2,3], which can save large amounts of energy if MXene coatings are applied to thermal equipment. In combination with small thickness, negligible weight per unit of area, and high flexibility, MXene coatings can provide an unprecedented level of thermal management in wearable or aerospace applications, where low weight of thermal protection is critical. For example, a 200-nm-

thick $Ti_3C_2T_x$ coating reaches an average IR emissivity of 0.06, comparable to polished metal[2]. Moreover, high electrical conductivity combined with the controlled IR emissivity allows the use of MXenes films, fibers or coatings as heaters for thermal management at low temperatures in space, at high altitudes, or in arctic climates[4,5].

MXene coating on polymers from aqueous suspension at room temperature can not only replace metallization performed by evaporating metal in vacuum, but also add numerous other functionalities. When integrated with a flexible polymer substrate, MXene is an excellent candidate for the thermal management of individuals and equipment with both Joule heating and thermal camouflage (minimal heat loss) capabilities. However, achieving a uniform assembly of MXene nanosheets to produce smooth coatings on many synthetic polymers from an aqueous suspension is challenging because of the hydrophobic and/or chemically inert nature of these polymers. These include many of the most important polymers such as polyethylene (PE), polyetheretherketone (PEEK), poly(tetrafluoroethylene) (PTFE),

[1]Hybrid Nano-Architectures and Advanced Manufacturing Laboratory, Department of Mechanical Engineering, Villanova University, Villanova, PA, USA. [2]A. J. Drexel Nanomaterials Institute and Department of Materials Science and Engineering, Drexel University, Philadelphia, PA, USA. [3]Department of Mechanical Engineering, Temple University, Philadelphia, PA, USA. [4]Electron Microscopy Center, Shared Equipment Authority, Rice University, Houston, TX, USA. [5]Department of Physics, Bryn Mawr College, Bryn Mawr, PA, USA. [6]These authors contributed equally: Liang Zhao, Lingyi Bi, Jiayue Hu. ✉e-mail: ling.liu@temple.edu; gogotsi@drexel.edu; bo.li@villanova.edu

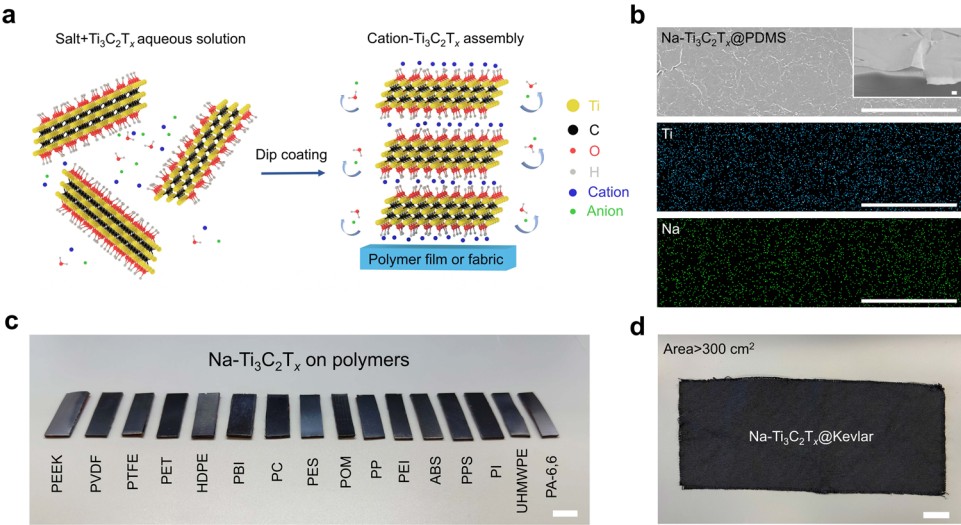

**Fig. 1 | Salt-assisted assembly of Ti$_3$C$_2$T$_x$ nanosheets on polymer substrates.**
**a** Schematic of Ti$_3$C$_2$T$_x$ nanosheets assembled on polymer substrates by SAA. **b** SEM images of the top surface of Na-Ti$_3$C$_2$T$_x$ nanosheets assembled on PDMS substrate and corresponding EDS mappings of Ti and Na elements. Inset is the tilted angle SEM image of the fractured cross-section. Scale bars, 5 μm. The thickness of Na-Ti$_3$C$_2$T$_x$ assemblies reaches 132 ± 40 nm, and the conductivity is up to 20,500 S cm$^{-1}$. **c** Digital images of Na-Ti$_3$C$_2$T$_x$ assemblies on different polymer films. Scale bar, 5 mm. **d** Digital image of large-scale Na-Ti$_3$C$_2$T$_x$ assemblies on Kevlar fabric. Scale bar, 3 cm.

and poly-paraphenylene terephthalamide (branded Kevlar) that have the best mechanical and thermal properties and are widely used in aerospace, high altitude, polar regions, and other extreme environments[6–9].

MXene nanosheets produced by wet chemical etching come with oxygen-based terminations and are hydrophilic in nature[10]. Take the most widely used MXene, Ti$_3$C$_2$T$_x$, as an example. T$_x$ represents surface terminations, mainly -OH, =O, and a small amount of -Cl and -F, providing Ti$_3$C$_2$T$_x$ nanosheets with a zeta potential below −30 mV and a pH of 6-7, and allowing them to form stable aqueous colloidal suspensions[11]. Assembly mechanisms of MXene nanosheets on polymers in aqueous suspension can be classified as forced deposition (driven by water evaporation) and self-assembly. In forced deposition, uniform coating usually requires wetting the polymer substrate with the MXene nanosheet colloidal suspension during fabrication, such as dip coating[12] or spray coating[13]. In self-assembly, chemical and/or physical interactions ensure the effective attraction between polymers and MXene nanosheets[3,14–16]. For example, polyelectrolyte can be introduced to create electrostatic attraction between the polymer and MXene nanosheets and among MXene nanosheets. Then, a layer-by-layer deposition of MXene nanosheets can be achieved[3,17]. Sometimes, the two mechanisms can be integrated to enhance the assembly effectiveness[18]. However, hydrophobic polymers (e.g., PE) cannot be wetted by MXene nanosheet aqueous suspension, and chemically inert polymers (e.g., Kevlar) do not bond to MXene nanosheets strongly enough to ensure adhesion of the MXene nanosheet. Establishing chemical bonds (ionic, hydrogen, and covalent bonds) between MXene nanosheets and polymers by adding adhesive polymer binders (e.g., polydopamine[15,16]), activating the polymer substrate by oxygen plasma or acid/base treatment[16], and creating electrostatic or hydrophobic interaction via the addition of surfactants (e.g., polyelectrolyte[3]) has been used to coat these polymers with MXene nanosheets. However, these approaches may jeopardize the performance of MXene coatings and/or the structural integrity of the polymer substrates. For example, the addition of poly(diallyldimethylammonium chloride) decreases the electrical conductivity of the resulting Ti$_3$C$_2$T$_x$ nanosheet film[3]. Plasma treatment may damage the surfaces of polymer substrates[14] and can be difficult to apply to some structures. Chemical treatments can be time-consuming and environmentally harmful. To address these challenges and include a wider range of polymers, a new MXene coating strategy is desirable. In this study, we report a non-destructive, efficient, and universal salt-assisted assembly (SAA) of MXene nanosheets from aqueous suspension on various polymer substrates by adding water-soluble salts into MXene nanosheet colloids (Fig. 1a).

## Salt-assisted assembly of MXene on polymers

We obtained hydrophilic Ti$_3$C$_2$T$_x$ nanosheets by etching Ti$_3$AlC$_2$ MAX phase and subsequent lithium-ion intercalation of the produced multilayer MXene (Supplementary Fig. 1)[19]. We chose hydrophobic polydimethylsiloxane (PDMS) as the substrate for demonstration because its molecular-level flat surface facilitates structural characterization (e.g., thickness and roughness) of the MXene coating. In an aqueous suspension, hydrophilic single- and few-layer Ti$_3$C$_2$T$_x$ nanosheets are stably dispersed as their negatively charged surface prevents aggregation of the nanosheets[20]. The SAA process includes adding salt (e.g., NaCl) to a 10 mg mL$^{-1}$ (or 1 wt. %) Ti$_3$C$_2$T$_x$ aqueous suspension, redispersion of the salt-added MXene suspension in an ultrasound bath (40 kHz, 60 W) for 15 min to prevent the aggregation of MXene nanosheets, and dipping a PDMS substrate into the redispersed suspension using a customized dip coater (Supplementary Fig. 2). The salt concentration can be tailored to control the assembly process. In this study, we kept the salt concentration at 0.01 mol L$^{-1}$ (0.058 wt.%) in the MXene suspension unless noted otherwise. A uniform coating of Ti$_3$C$_2$T$_x$ nanosheets on PDMS was produced (Fig. 1b). In contrast, dipping a PDMS substrate into pristine MXene suspension (without salt) using the same dipping parameters resulted in trace amounts of MXene on PDMS (Supplementary Fig. 3). The dip-coating process of SAA is much faster compared to conventional dip coating because of the differences in assembly mechanisms. In conventional dip coating, a thin layer of suspension containing particles wets the substrate withdrawn from the suspension, and the evaporation at the solid-liquid-vapor interface forces the deposition of particles onto the substrate. On the contrary, the SAA process is not evaporation-driven. The energetically favorable assembly happens at the MXene-polymer interface in the suspension. In the SAA process, the dipping speed reaches 1.5 m min$^{-1}$, 1–3 orders of magnitude higher than conventional dip coating, which is limited by slow evaporation.

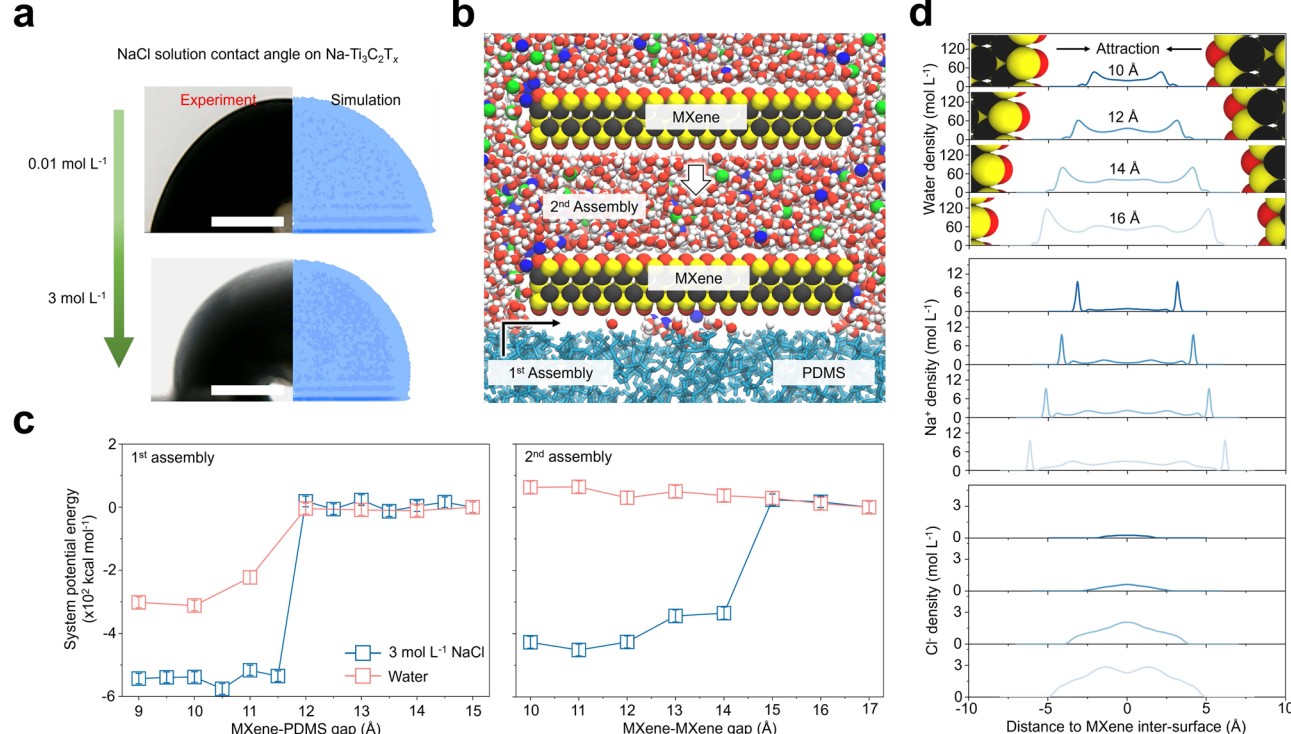

**Fig. 2 | Molecular dynamics simulation of salt-assisted assembly of MXene on PDMS. a** Contact angle (CA) of the 0.01 and 3 mol L$^{-1}$ NaCl solutions on Na-Ti$_3$C$_2$T$_x$ (left - experiment; right - MD simulation). Scale bars, 100 μm. **b** Illustration of the MD model (red - O; blue - Na$^+$; green - Cl$^-$; yellow - Ti; black - C; cyan - PDMS). **c** System potential energy variations during the 1$^{st}$ (MXene-PDMS) and 2$^{nd}$ (MXene-MXene) assembly processes in pure water and 3 mol L$^{-1}$ NaCl solution environment. The error bars represent the standard deviation calculated using energy outputs from the last three-quarters of the MD simulation runtime. **d** Evolution of the electric double layer (EDL) at four representative gap distances of the MXene nanosheets, where curves show the molar densities of water molecules (top), Na$^+$ ions (middle), and Cl$^-$ ions (bottom) in the presence of a 3 mol L$^{-1}$ NaCl solution. The Z-coordinate starts from the center of the nanosheets at every gap distance. The MXene models in the top panel show the MXene inter-surface locations for which the EDLs inside the MXene nanosheets are evaluated.

Herein, for chloride salt-assisted assembly of Ti$_3$C$_2$T$_x$, the cation was used to denote the samples. For example, Na-Ti$_3$C$_2$T$_x$ represents Ti$_3$C$_2$T$_x$ coating produced with NaCl. Salt ions are embedded in the assembled structures, as shown in the element mapping where both Na from salt and Ti from MXene are uniformly distributed across the entire surface (Fig. 1b). This low concentration of NaCl (0.01 mol L$^{-1}$) allows deposition of Ti$_3$C$_2$T$_x$ (after a 15-min dip coating and 132 ± 40 nm in thickness) with an electrical conductivity of ~20,500 S cm$^{-1}$, which is comparable to the best-reported values of Ti$_3$C$_2$T$_x$ films (Supplementary Table 2). Salt solutions with a higher concentration (up to saturated solution) can also be used, providing a process variable that can be used to control the assembly kinetics and the resultant MXene coating architecture (Supplementary Fig. 4).

SAA is a universal assembly method for hydrophobic and hydrophilic polymers. In Fig. 1c, we show Ti$_3$C$_2$T$_x$ coatings on 16 polymers, including those with the highest mechanical strength and thermal resistance, such as hydrophilic Kevlar and polyimide, and hydrophobic PE and PEEK (Supplementary Fig. 5). Before the adoption of SAA, many of these polymers needed complicated chemical modifications to be coated from aqueous MXene dispersions (Supplementary Table 3). However, with the utilization of SAA, all of them can be uniformly coated with Ti$_3$C$_2$T$_x$, as confirmed by SEM images (Supplementary Fig. 6). Furthermore, the thicknesses and electrical conductivities of the MXene coatings (Supplementary Table 4) are comparable to the those on PDMS. This suggests that the substrate chemistry does not affect the morphology and properties of the coating. Moreover, the SAA strategy is feasible for both flat and structured substrates. We have shown the MXene assembly on polymer fibers (Supplementary Fig. 7), curved surfaces, and 3D printed structures (Supplementary Fig. 8). Further, we prepared a large-scale (>300 cm$^2$) Kevlar fabric

coated with Ti$_3$C$_2$T$_x$ nanosheets (Fig. 1d), demonstrating the scalability of the SAA strategy.

## Mechanism of SAA strategy

The mechanism of SAA can be understood by analyzing the evolution of interactions between MXene, the substrate, and solution upon adding salt through both experiment and molecular dynamics (MD) simulation. In Fig. 2a, we have demonstrated increased contact angles (CA) of NaCl solution (i.e., from 0.01 mol L$^{-1}$ to 3 mol L$^{-1}$) on Na-Ti$_3$C$_2$T$_x$ thin film assembled on PDMS substrate using the SAA method. Similar trends are identified in a collection of polymer substrates (Supplementary Fig. 9). These results suggest that the salt solution repels both pristine MXene and polymer substrates, enabling energetically favorable adhesion of MXene on polymers. We have also found the CA of water on Na-Ti$_3$C$_2$T$_x$ thin film (74.1°) is higher than that on pristine Ti$_3$C$_2$T$_x$ thin film obtained by vacuum-assisted filtration (57.5°) suggesting salt treatment increases the hydrophobicity of MXene (Supplementary Fig. 9). In summary, both the dehydration effect of salt in the suspension and increased hydrophobicity of NaCl salt treated MXene promote the assembly of MXene. For NaCl concentration of 0.01 mol L$^{-1}$, the MD simulation yields a CA of 86.24 ± 1.35°, while the experimental measurement reaches 81.89 ± 3.42°. Upon increasing concentration to 3 mol L$^{-1}$, the MD-predicted CA increases to 95.81 ± 1.51°, while the experiment yields 92.48 ± 6.30°. The reasonably consistent results not only demonstrate the reliability of computation compared to the experiments but also collectively suggest a transition of MXene from being hydrophilic to more hydrophobic with the addition of salt. Together with the hydrophobic nature of PDMS, the ion-driven hydrophobicity increase assists the adherence of MXene to the PDMS substrate and subsequent MXene coating assembly.

The SAA was further studied computationally in two consecutive steps, i.e., the MXene-PDMS assembly and the MXene-MXene assembly (Fig. 2b). Figure 2c shows that, for MXene and PDMS, the system potential energy drops noticeably at a gap distance of approximately 12 Å, in both pure water and salty suspension. The presence of ions decreases the potential energy of the MXene surface, making assembly energetically more favored. These findings echo our experimental observations (Supplementary Fig. 3c) that while only a small number of MXene nanosheets adhere to PDMS in pure water, upon introducing salts, MXene nanosheets promptly adhere to and cover the entire PDMS surface. After the initial layer of MXene nanosheets has formed, ions continue to enable assemblies of multilayer MXene coatings. This phenomenon could be described by the extended DLVO (xDLVO) theory[21-24]. However, many system and material parameters must be determined before applying the xDLVO theory, which often requires extensive experiments and/or simulations. Alternatively, the MD simulation incorporates all significant physical factors, most of which were considered in the DLVO and xDLVO theories. The MD simulation is employed to compute the variations of the system potential as a metric to delineate the stability of the MXene coatings. In pure water, the system potential energy slightly increases when two MXene nanosheets approach each other, suggesting an energetically unfavored process that is unlikely to occur. However, in the NaCl solution (3 mol L$^{-1}$), the energy drops at a gap of approximately 15 Å, making the assembly energetically favored. Overall, ions assist assembly as they tune the hydrophobicity and mitigate the electronegativity and repulsion between MXene nanosheets.

During assembly, MXene nanosheets with adsorbed cations that deplete the negative surface charges undergo the expulsion of water molecules and anions, while some cations remain trapped within the assembled layers. The discharge process is depicted in Fig. 2d, which shows the evolution of electrical double layers (EDLs) between two MXene nanosheets that approach each other to mimic an assembly process. Molar densities of water molecules, cations, and anions are plotted in the MXene nanosheets of four different gap distances in the presence of a 3 mol L$^{-1}$ NaCl solution. At the gap distance of 16 Å, interfacial attraction leads to a peak water density of about 120 mol L$^{-1}$ in the first solvation shell (FSS), which doubles the density of bulk water (55.5 mol L$^{-1}$) (see Supplementary Fig. 10 for EDL on a MXene surface in bulk solution). As the assembly proceeds and the gap closes, water density in the confined FSS continues to drop. A similar reduction also occurs in the anions but not in the cations. A high density of cations exceeding three times the bulk concentration is found inside the nanosheets due to the electronegativity of MXene surfaces, and they remain trapped as water is depleted. By comparison, anions initially show a minor peak about 7 Å away from the MXene surface, gradually fading as the expulsion occurs.

It is important to note that adding salt to MXene suspension and MXene-polymer suspension can lead to flocculation or gelation of MXene (and polymer)[25-27], which may affect the assembly uniformity. In SAA, bath sonication is applied to redisperse MXene suspension after adding salt. The redispersed MXene nanosheets in the salt solution are stable during the assembly process, as demonstrated by their stable size distribution (Supplementary Table 5). The redispersed MXene nanosheets in their high energy states assemble on the polymer after the insertion of the polymer substrate to reduce the system's energy.

## Effects of the salt composition

The salt ions that adhere to the surface of MXene can affect the structures and properties of the final MXene assembly. In addition, the attached metal ions may enable new or enhanced functionalities, e.g., Ag$^+$ for antibacterial function[28], Al$^{3+}$ for water treatment[29], Sn$^{4+}$ for Li-ion batteries[30], and Pt$^{4+}$ for catalysis and electrocatalysis[31]. To fully explore the potential of the SAA method, we examined 49 salts with different combinations of cations and anions (Fig. 3a, b). The concentrations of the salts and Ti$_3$C$_2$T$_x$ nanosheets in the mixed suspension were kept constant at 0.01 mol L$^{-1}$ and 5 mg mL$^{-1}$, respectively. The assembly time was 15 minutes, and the substrate used was PDMS. The detailed morphologies of some assemblies can be found in Supplementary Fig. 11. Element mappings of these assemblies confirm that ions from the salt are attached to the surface of Ti$_3$C$_2$T$_x$ nanosheets (Supplementary Figs. 12 and 13). Mostly cations of the salt were found on the surface of Ti$_3$C$_2$T$_x$ nanosheets with a small number of anions. To prevent the interference of Cl on the surface of Ti$_3$C$_2$T$_x$ from the HF/HCl etching process, we used KBr as the salt, and EDS from scanning transmission electron microscope suggests the coexistence of K and Br (Supplementary Fig. 14). The same evidence can also be found in X-ray photoelectron spectra (XPS) of pristine Ti$_3$C$_2$T$_x$ and Cs-Ti$_3$C$_2$T$_x$ (Supplementary Fig. 15).

Salt species actively affect the assembly kinetics. For example, by fixing the anion (i.e., Cl$^-$) and changing the cations (i.e., Li$^+$, Na$^+$, K$^+$, Cs$^+$, Mg$^{2+}$, and Al$^{3+}$), we investigated the thickness and sheet resistance evolution of Ti$_3$C$_2$T$_x$ coatings on PDMS substrate with respect to assembly time and salt species (Fig. 3c, d). While similar trends of increased thickness and decreased sheet resistance with respect to assembly time were observed for all salts, under the same conditions, the Cs-Ti$_3$C$_2$T$_x$ coating was 10 times thicker than Na-Ti$_3$C$_2$T$_x$. The deposition speed can be tailored by the type of cations used, following the sequence of Cs$^+$ > Al$^{3+}$ > Mg$^{2+}$ > K$^+$ > Li$^+$ > Na$^+$. This trend can be attributed to the different dehydration capabilities of cations upon confinement in Ti$_3$C$_2$T$_x$ nanosheets[32], as well as the charge of the ion. Cosmotropic Al$^{3+}$ and Mg$^{2+}$ produce stronger electrostatic attraction when intercalated between MXene nanosheets[33]. It should be noted that though ions with higher dehydration capabilities, such as chaotropic Cs$^+$ and K$^+$, facilitate MXene assembly and lead to higher assembly speed, they result in increased coating roughness (Supplementary Fig. 16). These results revealed the ion-specific interactions (i.e., Hofmeister effect[34]) in salt-assisted assembly of MXene. Moreover, the addition of salt changes the spacing among MXene nanosheets. For example, by fixing the anion (i.e., Cl$^-$) and changing the cations (i.e., Li$^+$, Na$^+$), the spacing of stacked Ti$_3$C$_2$T$_x$ nanosheets changes from 14.1 (with Li$^+$) to 13.2 Å (with Na$^+$) (Supplementary Fig. 17) which can be used for tunable piezoresistive sensors, as previous studies have shown[35]. The Raman peak positions remain almost unchanged, independent of the ion used, indicative of no detectable chemical changes in Ti$_3$C$_2$T$_x$ coatings with metal ions compared to the pristine ones (Supplementary Fig. 18).

## Thermal management using MXene-coated polymers

To enable thermal management at high and low temperatures, Na-Ti$_3$C$_2$T$_x$ nanosheets were assembled on two of the most temperature-resistant polymers: PEEK film (Na-Ti$_3$C$_2$T$_x$@PEEK, coating thickness: ~170 nm) and Kevlar fabric (Na-Ti$_3$C$_2$T$_x$@Kevlar, coating thickness: ~870 nm) (Supplementary Fig. 19). The thermal management mechanism is shown in Fig. 4a. When a MXene coating is applied to the polymer sample placed on a hot plate, the low-emissivity MXene leads to the measured by the IR camera temperature ($T_{\text{reduction}}$) on the surface being much lower than the hot plate temperature ($T_{\text{radiation}}$). Figure 4b shows that when the hot plate was heated up to 300 °C for Na-Ti$_3$C$_2$T$_x$@PEEK and 400 °C for Na-Ti$_3$C$_2$T$_x$@Kevlar, the temperature difference ($T_{\text{radiation}}$ - $T_{\text{reduction}}$) reached ~200 °C for Na-Ti$_3$C$_2$T$_x$@PEEK and ~250 °C for Na-Ti$_3$C$_2$T$_x$@Kevlar. In comparison, the $T_{\text{radiation}}$ - $T_{\text{reduction}}$ of pure PEEK and Kevlar was only 14 °C and 58 °C, respectively (Supplementary Fig. 20). The stability of the thermal camouflage properties was examined over 50 heating and cooling cycles and in a long-term heating test for 48 hours, as shown in Figs. 4b and 4c (see detailed data in Supplementary Figs. 21–23 and Supplementary Table 6). The overlapping data at the 1$^{st}$, 25$^{th}$, and 50$^{th}$ cycles of

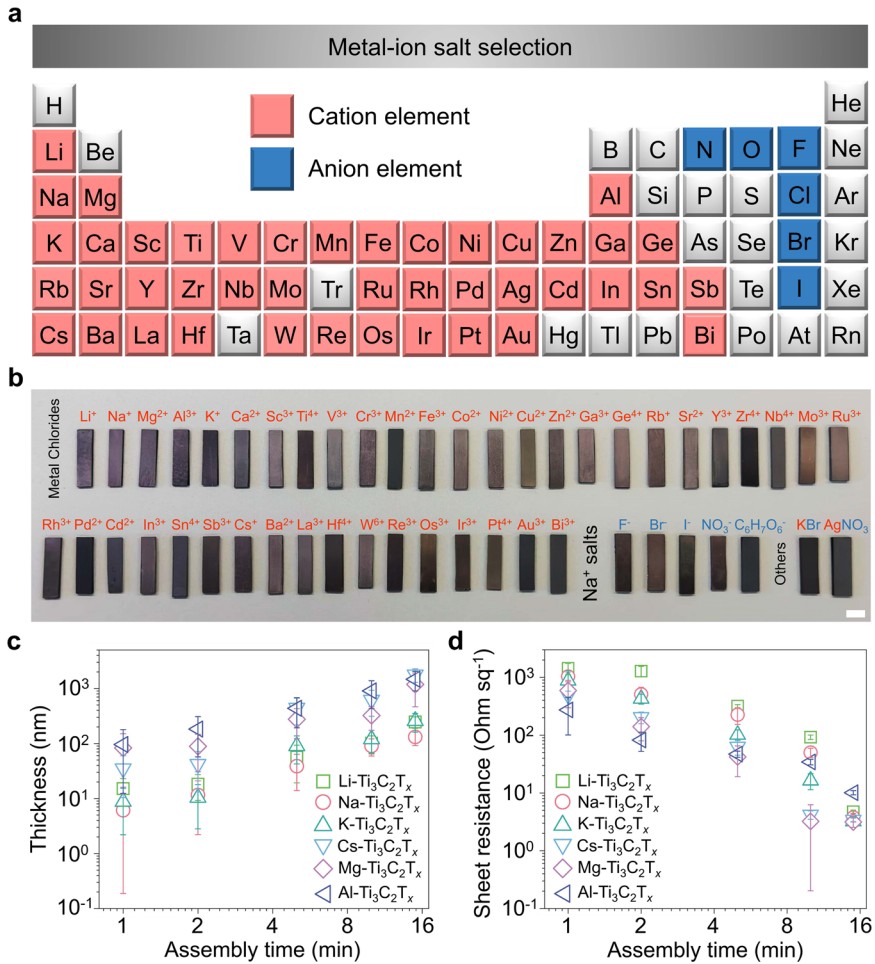

**Fig. 3 | Effects of cations and anions on salt-assisted assembly. a** Wide selection of cations and anions in the periodic table that can be used for SAA strategy. **b** Digital images of $Ti_3C_2T_x$ coatings on PDMS substrates deposited using diverse salts. Scale bar, 5 mm. **c** Thickness, and **d** sheet resistance evolution of $Ti_3C_2T_x$ coatings on PDMS deposited using different salts. The error bars are based on the standard deviation of 3 samples.

both sets of samples demonstrate excellent thermal stability and repeatability. After holding the sample at the highest $T_{radiation}$ for 48 h, $T_{reduction}$ reaches 117.2 °C for Na-$Ti_3C_2T_x$@PEEK and 158 °C for Na-$Ti_3C_2T_x$@Kevlar with only a small increase of 2.8 °C and 4.9 °C compared to their initial values, respectively. While the thermal camouflage capability can be mainly attributed to the Na-$Ti_3C_2T_x$ coating, the outstanding cyclic and long-term thermal camouflage stability is a result of the stable polymer substrate and the stable interface between the MXene and the polymer (Supplementary Fig. 24). In addition to the IR camera, we also used a thermocouple to probe the real surface temperature ($T_{real}$) of Na-$Ti_3C_2T_x$@Kevlar on the 400 °C hot plate (Supplementary Fig. 23). The real surface temperature can still reach 334.6 °C after 120 s under $T_{radiation}$ of 400 °C.

We further demonstrated the Joule heating performance of Na-$Ti_3C_2T_x$@Kevlar, as shown in Fig. 4d–f. By regulating the applied voltages, different heating temperatures (75.2 °C at 4 V and 192.9 °C at 8 V by IR camera) were rapidly achieved, where the $T_{real}$ measured by thermocouple showed similar values (71.7 °C at 4 V and 206.3 °C at 8 V) (Fig. 4e). Then, a long-term Joule heating test was performed at 4 V for 4 h (Fig. 4f), showing excellent stability. The stable performance can be attributed to the robustness of Na-$Ti_3C_2T_x$@Kevlar (Supplementary Fig. 24).

In wearable/flexible applications, both the flexibility and washing stability of MXene-coated polymer films and textiles are essential. We tested the bending durability of Na-$Ti_3C_2T_x$@Kevlar by comparing thermal camouflage performance and sheet resistance evolution

before and after 2000 bending cycles (Fig. 4g). Both $T_{reduction}$ (from 151.0 °C to 157.3 °C at $T_{radiation}$ = 400 °C) and sheet resistance (from 2.7 Ohm sq$^{-1}$ to 8.3 Ohm sq$^{-1}$ at room temperature) experience a small increase. We also examined the washing stability through a stirring washing test. The solution was stirred using a magnetic stir bar at 1000 rpm in a 1 L beaker to mimic the real washing conditions. Three types of solutions were tested: deionized water (DI) water, isopropanol (IPA) solution, and an industrial strength washing agent Synthrapol (10%, v/v). The sheet resistances of Na-$Ti_3C_2T_x$@PEEK and Na-$Ti_3C_2T_x$@Kevlar before and after washing were compared (Fig. 4h). After 168 hours of continuous washing in the harshest Synthrapol solution, sheet resistance increased from 3.4 Ohm sq$^{-1}$ to 82 Ohm sq$^{-1}$ for Na-$Ti_3C_2T_x$@PEEK and from 2.6 Ohm sq$^{-1}$ to 86.5 Ohm sq$^{-1}$ for Na-$Ti_3C_2T_x$@Kevlar. Considering a washing frequency of once a week for 1 h, such wearables can last for at least 3 years. The strong interface between Na-$Ti_3C_2T_x$ nanosheets and the polymer substrates can explain this excellent performance (Supplementary Figs. 25 and 26).

The combination of outstanding mid-IR reflectivity, low thermal conductivity, Joule heating capability, and bending and washing stability of MXene-coated high-performance polymers can be used in protective gear for individuals and equipment operating in extreme-temperature environments. As shown in Fig. 4i, we compared the performance range of the Na-$Ti_3C_2T_x$@Kevlar system (red region) with the state-of-the-art values from MXene@polymer systems (gray region) in terms of the highest $T_{radiation}$, the highest temperature difference in thermal camouflage, the highest Joule heating temperature,

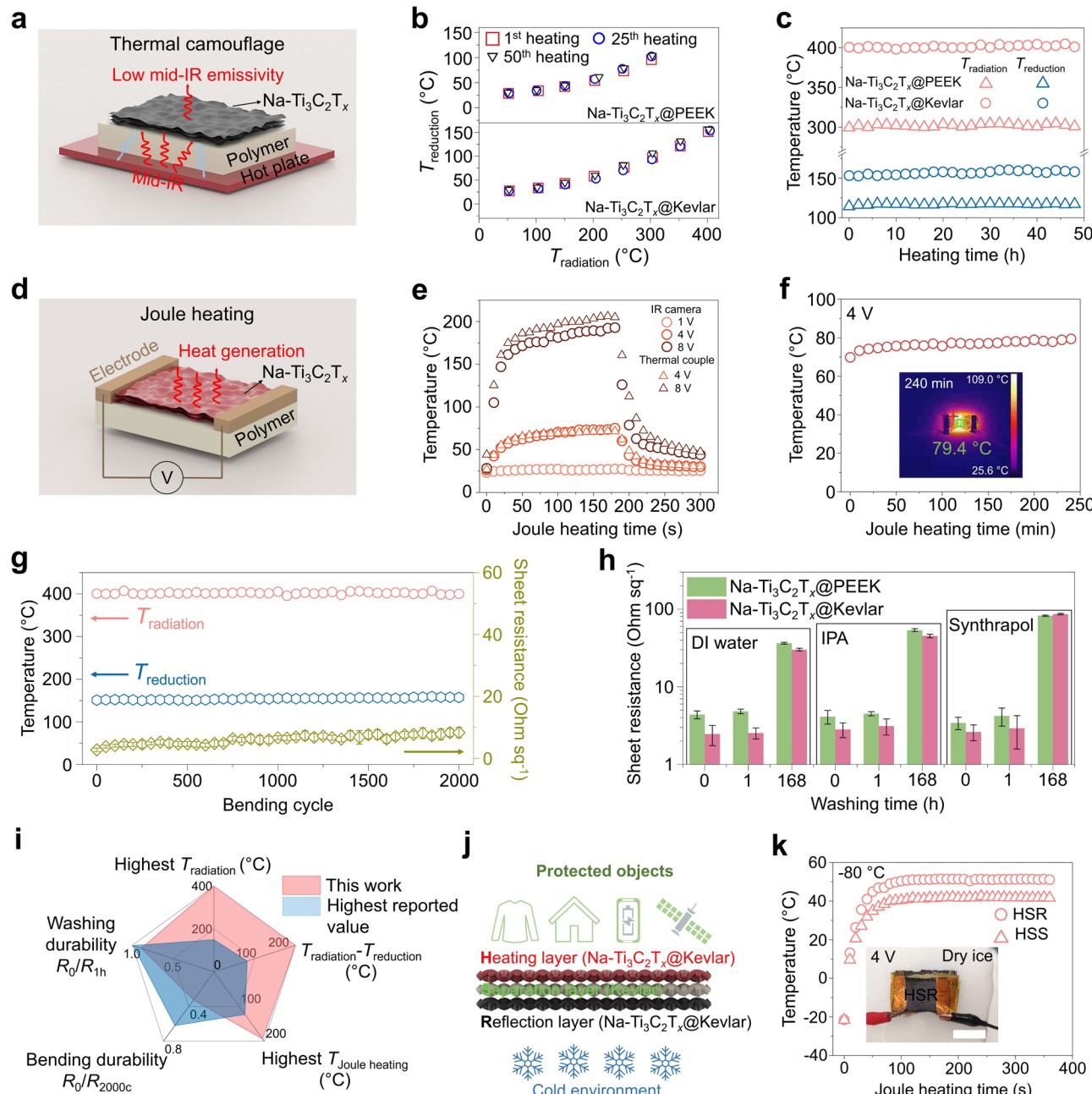

**Fig. 4 | Thermal management with Na-Ti$_3$C$_2$T$_x$ coatings on high-performance polymers. a** Schematic illustration of thermal camouflage process. **b** Evolution of reduction temperature ($T_{reduction}$) for Na-Ti$_3$C$_2$T$_x$@PEEK and Na-Ti$_3$C$_2$T$_x$@Kevlar within 50 heating cycles. **c** Evolutions of reduction temperature ($T_{reduction}$) for Na-Ti$_3$C$_2$T$_x$@PEEK and Na-Ti$_3$C$_2$T$_x$@Kevlar during 48 h at radiation temperatures ($T_{radiation}$) of 300 °C and 400 °C. **d** Schematic illustration of Joule heating. **e** Voltage-dependent Joule heating performance of Na-Ti$_3$C$_2$T$_x$@Kevlar. **f** Long-term Joule heating performance of Na-Ti$_3$C$_2$T$_x$@Kevlar under 4 V. **g** Evolution of reduction temperature at a $T_{radiation}$ of 400 °C and sheet resistance of Na-Ti$_3$C$_2$T$_x$@Kevlar during 2000 bending cycles. The error bars are based on the standard deviation of 10 samples. **h** Evolution of sheet resistance for Na-Ti$_3$C$_2$T$_x$@PEEK and Na-Ti$_3$C$_2$T$_x$@Kevlar after washing with DI water, IPA solution, and Synthrapol (10% in

volume) solution under 1000 rpm stirring. The error bars are based on the standard deviation of 5 samples. **i** Performance range of Na-Ti$_3$C$_2$T$_x$@Kevlar systems (highlighted in red) compared to existing state-of-the-art values from MXene@polymer systems. $R_0$ represents the initial sheet resistance of Na-Ti$_3$C$_2$T$_x$@Kevlar, $R_{1h}$ indicates resistance after 1 h washing, and $R_{2000c}$ is resistance after 2000 bending cycles[49–52]. **j** Schematic of a tri-layer H-S-R protection gear. From top to bottom, the H-S-R gear comprises a Heating layer (Na-Ti$_3$C$_2$T$_x$@Kevlar), a Separation layer (pure Kevlar), and a Reflection layer (Na-Ti$_3$C$_2$T$_x$@Kevlar). **k** The Joule heating temperatures of H-S-R protection gear in contact with dry ice (−78.5 °C). Scale bar, 5 mm. For comparison, the reference sample is H-S-S in which the reflection layer is substituted by a separation layer.

bending durability, and washing durability. Compared with other MXene composite structures with similar bending and washing durability, Na-Ti$_3$C$_2$T$_x$@Kevlar significantly outperforms them thermally (Supplementary Table 7). To demonstrate the potential applications, we designed a three-layer heat-management gear that can be used, e.g., in Mars exploration, to overcome the ultralow temperature (mean

temperature: −65 °C). As shown in Fig. 4j, from top to bottom, a Na-Ti$_3$C$_2$T$_x$@Kevlar layer was connected to an external power source and used as the heating layer (H-layer), an insulating Kevlar layer was used as a separator (S-layer) to prevent short circuits, and another Na-Ti$_3$C$_2$T$_x$@Kevlar layer was used to reflect mid-IR from the heating layer (R-layer) and prevent radiative loss to the dry ice (about −80 °C)

environment (Fig. 4k). Such gear can be called H-S-R gear. An IR camera was used to monitor the temperature of the H-layer. With a 4 V bias, the H-layer reached 51.1 °C. For comparison, if the R-layer is substituted with an S-layer to form an H-S-S gear, the H-layer can only reach 41.7 °C. This ~10 °C difference is significant considering that the Na-$Ti_3C_2T_x$ layer on the R-layer is only 872 nm thick. Note that it is not necessary to build H-S-R gear all over the body. As hands and feet are the most vulnerable body parts at low temperatures, we suggest building H-S-R into mittens and boots to reduce energy consumption. Of course, there is much room for optimization of this design. For example, other MXenes with higher emissivity can be used for the heating layer, and a smooth $Ti_3C_2T_x$ coating on a polymer film can be used for the reflective layer.

We developed a universal salt-assisted assembly protocol for fast large-scale assembly of MXene coatings on polymer substrates. By adding NaCl (or dozens of other salts) to the MXene colloidal suspension in water, we increased the hydrophobicity of both MXene and the polymer, promoting substrate-independent MXene deposition. Furthermore, the assembly kinetics, overall coating thickness, and architecture can be tailored by altering the salt ions and concentration. Coating high-performance polymers, such as Kevlar and PEEK, with MXenes holds the potential for significant advancements in thermal management under extremely low and high temperatures, preventing heat loss and protecting equipment and personnel. There are numerous applications for polymers coated with conductive MXene films, which possess a variety of optical and electronic properties. Incorporating catalytic metals like platinum or bactericidal silver ions further expands the range of potential applications of these materials.

## Methods

### Materials
All the materials used in this work are summarized in Supplementary Table S1.

### Synthesis of $Ti_3C_2T_x$ nanosheet suspension
$Ti_3C_2T_x$ was synthesized by the selective etching of $Ti_3AlC_2$ MAX phase powder (<40 μm particle size, Carbon-Ukraine) with a mixture of hydrofluoric (HF) (29 M, Acros Organics) and hydrochloric (HCl) (12 M, Fisher Chemical) acids[36]. First, 2 mL of HF, 12 mL of HCl, and 6 mL of de-ionized (DI) water were combined. After that, 1 g of MAX phase powder was added to the solution and stirred for 24 h at 35 °C. After etching, the reaction product was washed with DI water through 5-min centrifugation cycles at 2054 × $g$ until pH exceeded 6. The obtained sediment was dispersed in 20 mL of 1 mol L$^{-1}$ LiCl solution for Li$^+$ intercalation, and the reaction was allowed to proceed for 12–24 h at 300 rpm and 35 °C. The mixture was then washed with DI water to remove excess LiCl using 10-min centrifugation cycles at 2054 × $g$ until the supernatant darkened and the sediment swelled. Then a final washing cycle was performed at 2054 × $g$ for 1 h. The resulting clear supernatant was decanted and exchanged with DI water to redisperse the sediment with agitation. The mixture was centrifuged at 2054 × $g$ rpm for 10 min, with the dark supernatant being collected as a single layer $Ti_3C_2T_x$ dispersion. Sediment redispersion, 10-minute centrifugation at 2054 × $g$, and supernatant ($Ti_3C_2T_x$) collection were repeated till the supernatant became clear.

### Assembly of $Ti_3C_2T_x$ nanosheets on polymer substrates
A $Ti_3C_2T_x$ nanosheet colloidal suspension (10 mg L$^{-1}$, 10 mL) was diluted by adding a prepared salt solution (0.02 mol L$^{-1}$, 10 mL). In this way, the salt-$Ti_3C_2T_x$ suspension was fabricated, where the resultant salt and $Ti_3C_2T_x$ nanosheet concentrations were 0.01 mol L$^{-1}$ and 5 mg mL$^{-1}$, respectively. Because the $Ti_3C_2T_x$ nanosheets aggregation occurs when mixing salt and $Ti_3C_2T_x$ nanosheet suspension, the salt-$Ti_3C_2T_x$ suspension was sonicated for 15 min in a sonication bath (40 kHz, 60 W) to disperse $Ti_3C_2T_x$ nanosheet. After that, we used a customized dip

coater (average dipping speed = 1.5 m min$^{-1}$) to coat various polymer substrates. It is worth noting that the polymer substrates were submerged in the suspension during the whole assembly process. The assembled $Ti_3C_2T_x$ coatings on polymer substrates were dried with flowing compressed nitrogen gas to remove the excess suspension. To prevent salt crystals from precipitating from the suspension during drying, a DI water rinsing step was applied to the dried surface, followed by another round of nitrogen gas drying.

### Models and computational methods
Figure 2b illustrates the molecular model for energetically analyzing the assembly process. The model consisted of a polydimethylsiloxane (PDMS) layer serving as the substrate and MXene monolayers arranged parallel to the PDMS surface. The computational study involved two steps, i.e., the PDMS-MXene assembly, followed by the MXene-MXene assembly. The simulation box had the dimensions of 66 × 66 × 95 Å$^3$. Periodic boundary conditions were applied to all three dimensions. The amorphous PDMS comprised 20 chains, each with 50 repeating units. The PDMS model was initialized by energy minimization and relaxation to achieve the desired density with atomistic surface roughness. Each MXene monolayer consisted of 255-unit cells in the 17 × 15 pattern, resulting in a membrane of 45.63 Å × 44.78 Å positioned at the center of the simulation box. The $Ti_3C_2O_2$ MXene unit cell structure was derived from crystallographic experimental results[37], and the surface was negatively charged to reflect MXene's electro-negativity. The system was filled with 7338 water molecules and 3 mol L$^{-1}$ Na$^+$ and Cl$^-$ ions to study ionic effects. For comparison, another system with pure water was built, which had 7739 water molecules. In studying the PDMS-MXene assembly, the distance between the top surface of PDMS and the bottom surface of MXene varied from 15 Å to 9 Å. In studying the MXene-MXene assembly, the distance between the two MXene surfaces varied from 17 Å to 10 Å.

Molecular dynamics (MD) simulations were carried out using the large-scale atomic/ molecular massively parallel simulator (LAMMPS)[38]. Force field parameters for MXene were obtained from the work by Xu et al.[39,40], which is widely used to simulate the behavior of MXene in suspension environments[41,42]. Water molecules were described by the SPC/E model[43]. Ions were modeled by using the parameters proposed by Loche et al.[44]. PDMS was also described by the LJ potential and partial charges[45-47]. Detailed LJ parameters and charges are shown in Supplementary Table 8. Interatomic LJ interactions were described by the Lorentz-Berthelot combining rule. A cut-off distance of 10 Å was employed for both the LJ and Coulombic terms. Long-range interactions were handled by the particle-particle particle-mesh (PPPM) algorithm with a 10$^{-5}$ precision[48]. MD simulation was carried out under the NVT ensemble with a temperature of 293.15 K maintained using the Nose-Hoover thermostat and the atmospheric pressure. The time step was 2 fs. At each gap distance during the assembly analysis, the system was relaxed for 10 ns to reach equilibrium; subsequently, a production run of 15 ns was performed to obtain the number density distribution and averaged system potential energy.

### Characterization
The X-ray diffraction (XRD) analyses of $Ti_3AlC_2$ MAX phase powder, pristine $Ti_3C_2T_x$ film made by drop-casting on a glass slide, and $Ti_3C_2T_x$ assemblies on PDMS substrates obtained by SAA method were performed on a Rigaku Miniflex X-ray Diffractometer (40 kV and 15 mA) with Cu Kα radiation and a scanning speed of 10° min$^{-1}$. The $Ti_3C_2T_x$ nanosheet size distribution was measured by the dynamic light scattering (DLS) (Malvern Zetasizer Nano ZS) using a suspension diluted to 0.01 mg mL$^{-1}$. The monolayer $Ti_3C_2T_x$ nanosheet thickness on Si/SiO$_2$ wafer was determined by atomic force microscopy (AFM) (Park Systems NX10) in a noncontact mode. The contact angle of water and salt solutions (~5 μL) on polymer substrates and $Ti_3C_2T_x$ nanosheet films

was measured using a lab-made contact angle tester. The scanning electron microscope (SEM) images and x-ray energy dispersive spectrum (EDS) mapping of $Ti_3C_2T_x$ assemblies on different polymer films and fibers were acquired using a field emission scanning electron microscope (FE-SEM) (Hitachi S-4800 SEM) at 20 kV and 20 mA without sputtering. For the tilted angle view SEM images of the samples, we coated a 6 nm gold layer on both, the top surface and the side of the samples. The high angle annular dark field (HAADF) images, electron diffraction spectroscopy (EDS), and elemental mapping measurements were performed with double-corrected Titan cubed Themis G2 operated at 300 kV in the Electron Microscopy Center (EMC) of Shared Equipment Authority (SEA) at Rice University. The microscope is equipped with a Ceta camera, Gatan Quantum 966 energy filter, and an electron monochromator.

X-ray photoelectron spectra (XPS) were obtained using a PHI VersaProbe 5000 spectrometer (Physical Electronics, U.S.) with a monochromatic Al $K_\alpha$ X-ray source (1486.6 eV) at a 200 μm spot size and 50 W power. The spectra were collected with a 23.5 eV pass energy and an increment of 0.05 eV. All samples were mounted on conductive carbon tapes and electrically grounded via copper tape. High-resolution XPS data were fitted using the CasaXPS software package, employing a Tougaard background for transition metal-based species. The chemical states of $Ti_3C_2T_x$ MXene and the cations were deduced from core-level spectral fits. Raman spectra of $Ti_3C_2T_x$ and SAS $Ti_3C_2T_x$ coatings on polymer substrates were obtained using a WITec alpha300 confocal Raman microscope at an excitation laser wavelength of 785 nm with an 20× objective. The integration time was fixed to 2 s. The thickness and roughness of salt-treated $Ti_3C_2T_x$ assemblies on PDMS were measured by a Keyence VK-X1000 optical profilometer. The sample was placed under a 50× magnification lens of a Keyence VK-X1000 optical profilometer and evaluated using laser confocal scanning. Plane correction was performed on the scan using the accompanying software before three areas (with slight overlap and together covering the entire scan) were arbitrarily selected for roughness calculations. The areal average roughness (Sa) was chosen as the representative roughness parameter, which is calculated as the mean of the average height difference for the average surface. The sheet resistance of salt-treated $Ti_3C_2T_x$ assemblies was determined by four-point probe measurements (Jandel ResTest). For each sample, 10 points were measured, the average value was presented, and the standard deviation was calculated as the error. The surface temperature of Na-$Ti_3C_2T_x$ assemblies on PEEK film and Kevlar fabrics is recorded by an IR camera (HIKMICRO B20). The distance between the sample and the IR camera lens is fixed at 0.3 m, and the detected wavelength ranges from 8 to 14 μm. The absorbance/emissivity of salt-treated $Ti_3C_2T_x$ assemblies at different temperatures was tested using an FTIR spectrometer (Invenio-X, Bruker, Germany). An emission adapter (A540/3) was used to heat the samples and the black body reference (a soot layer on the metal sheet). The emissivity in the 5–25 μm range is given by the ratio of sample emission ($v$) and the reference emission at the same temperature ($T$).

## Reporting summary

Further information on research design is available in the Nature Portfolio Reporting Summary linked to this article.

## Data availability

All data are available in the main text or the supplementary information. Source data are provided with this paper.

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

## Acknowledgements

We thank Dr. A. Clark (Bryn Mawr College) for assistance with the XRD study, Dr. S. Dietrich (Villanova University) for the assistance with AFM, Drs. A. Jester and R. Lee (Villanova University) for the electrical conductivity measurements, and B. Hammill (Drexel University) for supporting MXene synthesis. L.Z., Y.L., A.F., and B.L. were supported in part by the U.S. National Science Foundation (Grants # AM-2003077, NRI-2221102, and MRI-2018852, B.L.), PA Manufacturing Fellows Initiative, Sport & Performance Engineering Seed Grant of College of Engineering, Villanova University. MXene synthesis by L.B., Y.G. and D.Z. at Drexel University was supported in part by the U.S. National Science Foundation (Grant # DMR-2041050, Y.G.). MD simulation by L.L. and J.H. was supported in part by the U.S. National Science Foundation (Grant # CBET-1751610, L.L.). The characterization of MXene films by T.Z. was supported by the U.S. Department of Energy, Office of Science, Office of Basic Energy Sciences (Grant # DE-SC0018618, Y.G.). Work at Bryn Mawr College was supported by the U.S. National Science Foundation (Grants # DMR-2242796, X.M.C.). The article processing charge is funded by Villanova University's College of Engineering and Falvey Memorial Library Scholarship Open Access Reserve Fund.

## Author contributions

L.Z. and B.L. conceived the concept of the assembly method and designed the experiment. Y.G. directed the MXene synthesis, experiment design, and data analysis. L.L. directed MD simulation. L.Z. performed the assembly process and sample preparation, SEM and EDS images, electrical conductivity test, FTIR, X-ray diffraction, contact angle measurements, thermal camouflage, and Joule heating test. L.B. synthesized MXene suspension and performed characterizations of MXene nanosheets and coating morphology. J.H. performed the MD simulation. D.Z. performed the high-temperature FTIR measurement. L.Z. and R.W. contributed to the Raman spectra measurement. G.G. performed the TEM imaging. Y.L. fabricated the 3D-printed PDMS substrates. A.F. collected and summarized the Joule heating data. T.Z. contributed to the XPS measurement. X.M.C. contributed to the XRD characterization of MXene coatings. L.Z., L.B., J.H., L.L., Y.G. and B.L. co-wrote the manuscript. All authors discussed the results and commented on the manuscript. L.Z., L.B. and J.H. contributed equally to this work.

## Competing interests

L.Z. and B.L. filed a U.S. Patent (Applicant: Villanova University; Application No. 2023/0286015 A1) based on the salt-assisted assembly

method. L.Z., L.B., Y.G., and B.L. filed a U.S. Provisional patent (Applicant: Villanova University; Application No. 63/623,093) based on thermal management applications. The remaining authors declare no competing interests.
