## [Transparent Peer Review file · Nature Communications]

Universal salt-assisted assembly of MXene from suspension on polymer substrates

Corresponding Author: Professor Bo Li

Version 0:

Reviewer comments:

Reviewer #1

(Remarks to the Author)

MXenes are novel materials attracting widespread contemporary interest due to their promise in various applications. The synthesis and structural characterization of these 2D materials have been comprehensively investigated in the recent past, while their use as building blocks in composites is a continuously growing area. Within this research direction, one of the important questions is the controlled assembly of MXenes into larger objects. This issue is addressed in the present paper NCOMMS-24-10363-T reporting that such an assembly can be induced by simple salts without the need of surface modification or other complicated experimental procedures. The topic is certainly hot and the results should attract considerable attention in the community. The manuscript is well-written, the figures are well-composed, while the conclusions are adequately supported by the experimental data obtained. I recommend the dissemination of this nice piece of work, however, please find below some questions to be answered prior publication.

(i) In Extended Data Fig. 1b and 1c, the DLS and AFM sizing is shown, while lateral dimension (assuming as the sphere diameter drawn around the MXene platelets) looks significantly larger in the AFM image compared to the reported 680 nm DLS data. Why? Could you provide size distributions based on AFM (or other microscopy technique) imaging and report the polydispersity indices determined by DLS and AFM?

(ii) It is clear from the manuscript that the salt-induced assembly takes place due to the combined effect of surface charge screening and ion specific adsorption of counterions on the MXene surface. These phenomena can be well described by the DLVO theory (10.1021/bk-2023-1457.ch002) and Hofmeister effect (10.1016/j.cocis.2016.05.005), respectively. Nevertheless, none of them mentioned in the manuscript.

(iii) On page 13, line 469, it is written that nanosheet aggregation occurs in the precursor MXene suspensions, while this was suppressed by sonication. Is there any direct experimental evidence for the absence of aggregates prior deposition to the polymer substrate? I am asking this because it is known that 2D nanosheets can assemble into lamellar structure and subsequently, into larger aggregates within relatively short time (see 10.1016/j.colcom.2021.100564).

(iv) Still regarding the colloidal properties of the MXene dispersions, the authors quote reference 22 on page 3, line 97 that the nanosheets are stable in aqueous dispersions due to high surface charge. Is it also true in the presence of multivalent counterions, e.g., aluminum(III) in the present study, which can induce aggregation at low concentration (see Schulze-Hardy rule in 10.1351/pac198052051151 or recently in 10.1007/s00396-020-04665-w)?

(v) In Fig. 3c, more than an order of magnitude difference can be seen within the thicknesses obtained in different salt solutions. The trend looks systematic at each assembly time and the presence of multivalent cations leads to larger thicknesses. Please comment on this. Finally, what does "strong interaction" exactly mean on page 8, line 291? Is this also salt specific? Its origin is electrostatic, van der Waals, or something else?

Reviewer #2

(Remarks to the Author)

In the manuscript titled "Universal salt-assisted assembly of MXene from solution on polymers," Zhao and coworkers use aqueous salt solutions to "hydrophobize" Ti₃C₂T_x nanosheets so that they deposit on polymer films. Experimental work is complemented by MD simulations and the application of these coatings in thermal management is reported. Overall this work is scientifically sound and of broad interest to researchers in composites, thermal management, and technology development. However, a number of issues must be addressed. These primarily focus on the accessibility of the work to readers and materials characterization.

Major Comments:

- In the background, LbL assembly of MXenes and polymers should be briefly addressed (paragraph on lines 66-89 could possibly be split into two in this case)
- Critically missing from the background is discussion of the work of Cao and coworkers on the use of aqueous solutions of

salt to flocculated MXenes and use these as surfactants and couple them with polymerizations to give polymer-MXene composites. See, for example: Langmuir, 2021, 37, 2649; ACS Applied Materials & Interfaces, 2021, 13, 51556; 2D Materials, 2022, 9, 044004.

- The first paragraph would benefit of specific examples, rather than just claiming, e.g., “unprecedented level of thermal management”. In introduction, it is also confusing when “MXene” is used alone, and it would help if nanosheets, coatings, etc. are differentiated (e.g., paragraph 3).
- When SAA is first introduced, more detail is needed. Given the reported SAA is not a standard dip coating, it should be described in the main text (calling it dip coating indicates an evaporation driven mechanism, which is not happening, as described in caption of ED Fig 2). Is “1.524 m/min” the correct number of significant figures? (Line 100). Can the solution be described in terms of wt% MXene nanosheet and wt% salt? Some of such information that would be helpful to readers is not explained until line 204 and beyond.
- Paragraph starting line 227, the authors state assembly time changes, but then talk about thickness. If assembly rate is to be discussed, thickness should be measured over time. In general, can the authors add general comments about how the time was selected? It seems that the conditions were judiciously selected, but can the authors provide any insight into a minimum or maximum time for coating? Or a relationship between thickness and time for each?

Minor Comments:

- The title with “on polymers” is confusing. It might be better to say “on polymer substrates”
- In abstract, rather than claim the salt neutralizes the charge, it should state something along the lines of electrostatically hydrophobized (the nanosheets themselves still carry a charge)
- Line 100-101, what is the definition of “poor assembly”.
- Paragraph starting line 188, could the term depletion be used here?
- Is the trend in spacing between the nanosheets consistent with cation radius? If not, why not?
- In Fig. 3, are the error bars between different substrate samples or within a single substrate? If the latter, does this indicate roughness?
- What’s the largest scale sample that has been coated? Given that with addition of salt the nanosheets aggregate and needs to be redispersed (as discussed in the experimental), is there a timeframe within which the solution needs to be used? What are the limitations of the processing approach?
- ED Fig 3, can the height profile be changed to give meaningful data to most of the image (the one red spot makes the blue/green non-differentiable)
- ED Fig. 6 is referenced as showing “uniform coating”, however these are only electron micrographs and don’t show a large area; is there a better technique to complement SEM to show uniform coating?
- ED Fig. 7, it would be helpful to show the non-coated fabrics and fibers to highlight the impact of nanosheet assembly. This same comment can be applied to ED Fig. 8 (with just SEM images it’s difficult to confirm/support that the nanosheets have been coated)
- For the different cations, is there any elemental analysis (such as at% by EDS) that can be used to evaluate composition). Is there a correlation between charge and composition?
- From the FTIR spectra of ED Fig. 15, comparison of the -OH stretching frequency intensity is tenuous given changes across different peaks. It would be more reliable if another peak could be used as an internal standard, or this discussion could be softened (or removed)
- ED Fig. 17, state how roughness was determined

Version 1:

Reviewer comments:

Reviewer #1

(Remarks to the Author)

The authors addressed my comments, the paper can published as is.

Reviewer #3

(Remarks to the Author)

The manuscript presents a highly innovative study on the application of MXene coatings, specifically Na-Ti₃C₂T_x, on high-performance polymers for thermal management in extreme environments. The authors demonstrate a novel, salt-assisted assembly method for achieving uniform, ultra-thin MXene coatings on diverse polymer surfaces, addressing significant challenges in the deposition of MXenes on hydrophobic and inert surfaces. The results showcase the remarkable potential of MXene coatings in various applications, such as radiation temperature management and Joule heating for harsh environments.

The salt-assisted assembly method described is a significant advancement in MXene coating technology. The use of salts to deplete surface charges and enable electrostatic deposition of MXene on polymer surfaces is both novel and effective. This method offers a universal approach to creating mechanically stable and washable MXene coatings, expanding the range of materials that can be coated with MXenes. I like the idea of using a wide range of salts for testing.

The study provides a thorough characterization of the MXene coatings, including their thermal management capabilities, mechanical stability, and washability. The results demonstrate that the coatings are highly stable under extreme conditions, with excellent performance after multiple heating cycles, bending, and washing. This level of detail significantly enhances the credibility of the findings.

It highlights the practical applications of the MXene coatings in extreme environments, such as outer space and polar regions. The demonstration of a ~ 200 °C radiation temperature drop on a 300 °C PEEK substrate and the effective Joule heating of Kevlar fabric at -80 °C using a 4V battery shows the practical potential of these coatings in real-world scenarios. This broadens the impact of the research.

The inclusion of molecular dynamics simulations provides a deeper understanding of the underlying mechanisms of MXene assembly and performance. The simulations suggest that the salt-induced changes in interlayer spacing, water expulsion, and ion trapping play a crucial role in the performance of the coatings, offering valuable insights that complement the experimental data.

However, before publication, I recommend double check and update the discussion with works focused on layer-by-layer assembly of MXenes, especially to polymer substrates, such as, e.g., <https://onlinelibrary.wiley.com/doi/full/10.1002/smt.202201252> (textile fabric fibers). This is just a suggestion but could increase the understanding of readers in terms of potential impact of the paper.

The manuscript effectively covers a wide range of performance parameters, including thermal management, Joule heating, and mechanical stability under various conditions. The analysis of different salts to tailor the assembly kinetics and morphology of the coatings adds a valuable dimension to the study, showcasing the versatility of the proposed approach. Therefore, I strongly recommend the manuscript for publication after minor revision. The findings have the potential to significantly impact the processing and development of advanced layers for thermal management and protective applications in extreme conditions.

Point-by-Point Response

Reviewer #1 (Remarks to the Author):

MXenes are novel materials attracting widespread contemporary interest due to their promise in various applications. The synthesis and structural characterization of these 2D materials have been comprehensively investigated in the recent past, while their use as building blocks in composites is a continuously growing area. Within this research direction, one of the important questions is the controlled assembly of MXenes into larger objects. This issue is addressed in the present paper NCOMMS-24-10363-T reporting that such an assembly can be induced by simple salts without the need of surface modification or other complicated experimental procedures. The topic is certainly hot and the results should attract considerable attention in the community. The manuscript is well-written, the figures are well-composed, while the conclusions are adequately supported by the experimental data obtained. I recommend the dissemination of this nice piece of work, however, please find below some questions to be answered prior publication.

(1) In Extended Data Fig. 1b and 1c, the DLS and AFM sizing is shown, while lateral dimension (assuming as the sphere diameter drawn around the MXene platelets) looks significantly larger in the AFM image compared to the reported 680 nm DLS data. Why? Could you provide size distributions based on AFM (or other microscopy technique) imaging and report the polydispersity indices determined by DLS and AFM?

Response: We appreciate the careful examination by the reviewer. We double-checked the average size of MXene nanosheets by DLS. The Z average was 1078 ± 84 nm with a polydispersity value (PDI) of 0.006. A correction has been made accordingly in the manuscript. Correlations between the Z-average measured by DLS and flake dimensions measured by SEM have been reported on page 60 of Kathleen Maleski's PhD thesis (2020, Drexel University). It was observed that most MXene flakes can be approximated as rectangles with an aspect ratio (length/width) of 1.6/1, where the width is most similar to the Z-average. This is in good agreement with the SEM images of the MXene nanosheets, where the average length of 50 MXene nanosheets was calculated to 1612 ± 588 nm (PDI=0.133). A representative SEM image was added to ED Fig. 1 as shown below.

Extended Data Fig. 1 | Characterization of pristine $\text{Ti}_3\text{C}_2\text{T}_x$ nanosheets. **a**, X-ray diffraction (XRD) patterns of Ti_3AlC_2 MAX phase powder and $\text{Ti}_3\text{C}_2\text{T}_x$ nanosheet film. The intensity of (002) peak for $\text{Ti}_3\text{C}_2\text{T}_x$ nanosheet film is enhanced and the position shifts towards lower 2θ compared with Ti_3AlC_2 MAX phase powder, demonstrating the successful etching and delamination¹. The $\text{Ti}_3\text{C}_2\text{T}_x$ nanosheet film for XRD measurement is prepared by drop-casting method on a glass slide. **b**, Representative dynamic light scattering (DLS) curve of $\text{Ti}_3\text{C}_2\text{T}_x$ nanosheet solution (concentration: 0.01 mg mL^{-1}). The calculated average lateral size of $\text{Ti}_3\text{C}_2\text{T}_x$ nanosheets is $\sim 1078 \text{ nm}$. **c**, Representative SEM image of monolayer $\text{Ti}_3\text{C}_2\text{T}_x$ nanosheet. Scale bar, $1 \mu\text{m}$. **d**, Atomic force microscope (AFM) image of a monolayer $\text{Ti}_3\text{C}_2\text{T}_x$ nanosheet. Scale bar, 500 nm . **e**, Height profiles in (d). The thickness of the monolayer $\text{Ti}_3\text{C}_2\text{T}_x$ nanosheet is $\sim 1.8 \text{ nm}$ due to water and other adsorbed species between the flake and the substrate, which is similar to the reported values for single-layer MXene flakes².

(2) It is clear from the manuscript that the salt-induced assembly takes place due to the combined effect of surface charge screening and ion specific adsorption of counterions on the MXene surface. These phenomena can be well described by the DLVO theory (10.1021/bk-2023-1457.ch002) and Hofmeister effect (10.1016/j.cocis.2016.05.005), respectively. Nevertheless, none of them mentioned in the manuscript.

Response: We thank the reviewer for this constructive comment. We agree that the extended DLVO (i.e., xDLVO) theory can be used to describe the phenomena of salt-assisted assembly. The xDLVO theory, compared to the classical DLVO theory, includes additional contributions from hydration (repulsive) and hydrophobic (attractive) interactions, thereby accounting for major interactions during the assembly process of solid surfaces in salt solutions. However, the xDLVO theory assumes continuous medium and homogeneous surfaces with low charge densities, and is often challenged by various size effects and the nanoconfinement effect. There are also many system and material parameters that must be determined before the theory may be applied to specific

assembly processes, which often require extensive experiments and/or simulations (e.g., measuring contact angle to determine surface tension parameters).

Alternatively, the molecular dynamics (MD) simulation incorporates all factors that are significant to study salt-assisted assembly processes, most of which are considered in the DLVO theories. With the native capability of studying atomic confining effects all the way down to the sub-nanoscale, MD accurately predicts key quantities including the binding energy, the pull-off force, and the primary minimum. It can also reveal ion-specific density distributions and the electric double layer (EDL) evolution, together with the energy evolution. As such, this study used MD to reveal assembly energetics and structures.

The Hofmeister effect refers to the phenomenon where the ion species in the solution influence the solubility, stability, and behavior of colloids, proteins, and other macromolecules. The resulting Hofmeister series is highly case-dependent. To reveal the Hofmeister effect in the salt-assisted assembly of MXene, we conducted a series of experiments and investigated ion-specific interactions, as shown in Fig. 3. The results indicate the capabilities of different ions in assisting MXene assembly.

Manuscript modification:

line 183-189: After the initial layer of MXene nanosheets has formed, ions continue to enable assemblies of multilayer MXene coatings. This phenomenon could be described by the extended DLVO (xDLVO) theory (*One Hundred Years of Colloid Symposia: Looking Back and Looking Forward. American Chemical Society*, **1457**, 31-47 (2023); *Interf. Sci. Technol.* **16**, 31-48 (2008); *Water Research* **171**, 115401 (2020); *Langmuir* **32**, 88-101 (2016)). However, many system and material parameters must be determined before applying the xDLVO theory, which often requires extensive experiments and/or simulations. Alternatively, the MD simulation incorporates all significant physical factors, most of which were considered in the DLVO and xDLVO theories. The MD simulation is employed to compute the variations of the system potential as a metric to delineate the stability of the MXene coatings. In pure water, the system potential energy slightly increases when two MXene nanosheets approach each other, suggesting an energetically unfavored process that is unlikely to occur.

line 258-259: Those results revealed the ion-specific interactions (i.e., Hofmeister effect (*Curr. Opin. Colloid Interf. Sci.* **23**, 41-49 (2016))) in salt-assisted assembly of MXene flakes.

(3) On page 13, line 469, it is written that nanosheet aggregation occurs in the precursor MXene suspensions, while this was suppressed by sonication. Is there any direct experimental evidence for the absence of aggregates prior deposition to the polymer substrate? I am asking this because it is known that 2D nanosheets can assemble into lamellar structure and subsequently, into larger aggregates within relatively short time (see 10.1016/j.colcom.2021.100564).

Response: To address this concern, we have performed new DLS experiments to compare the aggregation status, as shown in the attached table. We have compared DLS under three conditions: (1) pristine $Ti_3C_2T_x$ solution (well-dispersed solution without aggregation), (2) $Ti_3C_2T_x$ solution after the addition of 0.01 M NaCl (the sonication was not turned on to redisperse the $Ti_3C_2T_x$ solution), (3) redispersed $Ti_3C_2T_x$ solution using sonication after the addition of 0.01 M NaCl.

For Condition 1, the average size is 1078 nm. For Condition 2, the value increases to 1265 nm. Comparing Conditions 1 and 2 suggests that aggregation happens if we do not use sonication to redisperse the solution.

For Condition 3, we used sonication to redisperse the $Ti_3C_2T_x$ solution with 0.01 M NaCl for 15 min. This is exactly the process we perform before the assembly on the polymer substrates. In Condition 3, we keep monitoring the aggregation status of the solution after turning off the sonication for 15 min. The DLS results show an average size of 1093 nm right after the sonication and 1089 nm 15 min later.

These results address the reviewer's concerns. First, adding salt without sonication will lead to aggregation. Second, sonication can redisperse MXene solution and prevent aggregation. Third, after sonication redispersion, the solution can remain in its dispersed status during the assembly process, which takes up to 15 minutes for our samples.

	Pure $Ti_3C_2T_x$	$Ti_3C_2T_x$ with 0.01 M NaCl (no sonication)	$Ti_3C_2T_x$ with 0.01 M NaCl (15 min sonication)	$Ti_3C_2T_x$ with 0.01 M NaCl (15 min sonication and wait for 15 min)
Size (nm)	1078±84	1265±98	1093±91	1089±176

(4) Still regarding the colloidal properties of the MXene dispersions, the authors quote reference 22 on page 3, line 97 that the nanosheets are stable in aqueous dispersions due to high surface charge. Is it also true in the presence of multivalent counterions, e.g., aluminum(III) in the present study, which can induce aggregation at low concentration (see Schulze-Hardy rule in 10.1351/pac198052051151 or recently in 10.1007/s00396-020-04665-w)?

Response: As the reviewer has correctly pointed out, the multivalent ions, e.g., Al^{3+} , induce aggregation. However, when sonication is used, the problem of aggregation is alleviated, as evident from the DLS values reported below. Though after the immediate addition of 0.01 M $AlCl_3$, the MXene significantly agglomerated to 15567 nm, a 15-minute sonication was sufficient in breaking the agglomerates down to 1135 nm, close to the pristine state.

	Pure $Ti_3C_2T_x$	$Ti_3C_2T_x$ with 0.01 M $AlCl_3$ (no sonication)	$Ti_3C_2T_x$ with 0.01 M $AlCl_3$ (15 min sonication)	$Ti_3C_2T_x$ with 0.01 M $AlCl_3$ (15 min sonication and wait for 15 min)
Size (nm)	1078±84	15567±3673	1135±147	1169±261

For the DLS data in Comments (3) and (4), we added them as ED Table 5.

(5) In Fig. 3c, more than an order of magnitude difference can be seen within the thicknesses obtained in different salt solutions. The trend looks systematic at each assembly time and the presence of multivalent cations leads to larger thicknesses. Please comment on this. Finally, what does “strong interaction” exactly mean on page 8, line 291? Is this also salt specific? Its origin is electrostatic, van der Waals, or something else?

Response: Yes, the deposition speed can be tailored by the type of cations used, following the sequence of $\text{Cs}^+ > \text{Al}^{3+} > \text{Mg}^{2+} > \text{K}^+ > \text{Li}^+ > \text{Na}^+$. This trend can be attributed to different dehydration capabilities of cations upon confinement in $\text{Ti}_3\text{C}_2\text{T}_x$ nanosheets (*Gao et al. Energy & Environmental Science, 13(8), 2549-2558*), as well as the charge of the ion. Cosmotropic Al^{3+} and Mg^{2+} produce stronger electrostatic attraction when intercalated between MXene nanosheets (*Shpigel et al. Journal of the American Chemical Society 140.28 (2018): 8910-8917*). It should be noted that ions with higher dehydration capabilities, such as chaotropic Cs^+ , facilitate the MXene assembly and lead to higher assembly speed.

The strong interactions here include two parts: between MXene nanosheets and between MXene nanosheets and polymer substrates. Their origin of both is attributed to electrostatic attraction, which occurs due to the depletion of surface charges. This depletion can be facilitated by the addition of salts.

Reviewer #2 (Remarks to the Author):

In the manuscript titled “Universal salt-assisted assembly of MXene from solution on polymers,” Zhao and coworkers use aqueous salt solutions to “hydrophobize” $\text{Ti}_3\text{C}_2\text{T}_x$ nanosheets so that they deposit on polymer films. Experimental work is complemented by MD simulations and the application of these coatings in thermal management is reported. Overall this work is scientifically sound and of broad interest to researchers in composites, thermal management, and technology development. However, a number of issues must be addressed. These primarily focus on the accessibility of the work to readers and materials characterization.

Major Comments:

(1) In the background, LbL assembly of MXenes and polymers should be briefly addressed (paragraph on lines 66-89 could possibly be split into two in this case)

Response: We have added LBL assembly in the summary of the self-assembly method.

Line 75 to 78: In self-assembly, chemical and/or physical interactions ensure the effective attraction between polymers and MXene nanosheets^{2,3,16,17}. For example, polyelectrolyte can be introduced to create electrostatic attraction between polymer and MXene nanosheets and among MXene nanosheets. Then, a layer-by-layer deposition of MXene nanosheets can be achieved³.

(2) Critically missing from the background is discussion of the work of Cao and coworkers on the use of aqueous solutions of salt to flocculated MXenes and use these as surfactants and couple them with polymerizations to give polymer-MXene composites. See, for example: *Langmuir*, 2021, 37, 2649; *ACS Applied Materials & Interfaces*, 2021, 13, 51556; *2D Materials*, 2022, 9, 044004.

Response: We added the necessary discussion.

Line 223 to 229: It is important to note that adding salt to MXene solution and MXene-polymer solution can lead to flocculation or gelation of MXene (and polymer)²⁵⁻²⁷ which may affect the assembly uniformity. In SAA, bath sonication is applied to redisperse MXene solution after adding salt. The redispersed MXene nanosheets in the salt solution are stable during the assembly process, as demonstrated by their stable size distribution (Extended Data Table 5). The redispersed MXene

nanosheets in their high energy states assemble on the polymer after insertion of the polymer substrate to reduce the system energy.

(3) The first paragraph would benefit of specific examples, rather than just claiming, e.g., “unprecedented level of thermal management”. In introduction, it is also confusing when “MXene” is used alone, and it would help if nanosheets, coatings, etc. are differentiated (e.g., paragraph 3).

Response: We added an example of the thermal management of MXene coating, i.e., thermal camouflage.

Line 52-53: For example, a 200-nm-thick $\text{Ti}_3\text{C}_2\text{T}_x$ coating reaches an average IR emissivity of 0.06, which is comparable to polished metal⁴.

We have modified “MXene” to “MXene nanosheets”.

(4) When SAA is first introduced, more detail is needed. Given the reported SAA is not a standard dip coating, it should be described in the main text (calling it dip coating indicates an evaporation driven mechanism, which is not happening, as described in the caption of ED Fig 2). Is “1.524 m/min” the correct number of significant figures? (Line 100). Can the solution be described in terms of wt% MXene nanosheet and wt% salt? Some of such information that would be helpful to readers is not explained until line 204 and beyond.

Response: Exactly. The dip coating process we used is different from the conventional one. We have added the description (lines 104 to 119):

“SAA process includes adding salt (e.g., NaCl) to 10 mg mL⁻¹ (or 1 wt. %) $\text{Ti}_3\text{C}_2\text{T}_x$ aqueous solution, redispersing the salt-added MXene solution in an ultrasound bath (40 kHz, 60W) for 15 min to prevent the aggregation of MXene nanosheet, and dipping PDMS substrate into redispersed solution using a customized dip coater (Extended Data Fig. 2). The salt concentration can be tailored to control the assembly process. In this study, we kept the salt concentration at 0.01 mol L⁻¹ (0.058 wt.%) in the MXene solution unless noted otherwise. A uniform coating of $\text{Ti}_3\text{C}_2\text{T}_x$ nanosheets on PDMS was produced (Fig. 1b). In contrast, dipping PDMS substrate into pristine MXene solution (without salt) using the same dipping parameters resulted in trace amounts of MXene on PDMS (Extended Data Fig. 3). The dip-coating process of SAA is much faster compared to conventional dip coating because of the differences in assembly mechanisms. In a conventional dip coating, a thin layer of solution containing particles wets the substrate withdrawn from the solution, and the evaporation at the solid-liquid-vapor interface forces the deposition of particles onto the substrate. In opposite, the SAA process is not evaporation-driven. The energetically favorable assembly happens at the MXene-polymer interface in the salt solution. In the SAA process, the dipping speed reaches 1.5 m/min, which is 1-3 orders of magnitude higher than conventional dip coating, which is limited by slow evaporation.”

We changed the “1.524 m/min” to “1.5 m/min”.

We calculated the weight percentage of MXene and salt. For 10 mg/mL MXene, it is 1 wt. %. For 0.01 M NaCl salt solution, it is 0.058 wt.%. We also added the information in the main manuscript (line 104 and line 109).

(5) Paragraph starting line 227, the authors state assembly time changes, but then talk about thickness. If assembly rate is to be discussed, thickness should be measured over time. In general,

can the authors add general comments about how the time was selected? It seems that the conditions were judiciously selected, but can the authors provide any insight into a minimum or maximum time for coating? Or a relationship between thickness and time for each?

Response:

We observed an increase in coating thickness with the increase in assembly time from 1 minute to 15 minutes, as reported in Figure 3c. It is possible that this trend will continue for even longer assembly times of over 15 minutes. However, although the resulting lower resistance from a longer assembly is favorable for thermal management (Figure 3d), longer assembly times negatively affect process efficiency. Therefore, we have selected a 15-minute assembly time for the parametric studies.

Minor Comments:

(6) The title with “on polymers” is confusing. It might be better to say “on polymer substrates”

Response: We have changed the title to “Universal salt-assisted assembly of MXene from solution on polymer substrates”.

(7) In abstract, rather than claim the salt neutralizes the charge, it should state something along the lines of electrostatically hydrophobized (the nanosheets themselves still carry a charge)

Response: We modified the sentence “The salt added to the $Ti_3C_2T_x$ aqueous colloid neutralizes MXene’s surface charge and deposits MXene onto the polymer surface” to “The salt added to the $Ti_3C_2T_x$ aqueous colloid depletes MXene’s surface charges and deposits electrostatically hydrophobized MXene onto the polymer surface.”

(8) Line 100-101, what is the definition of “poor assembly”.

Response: “Poor assembly” indicates that within a certain assembly time, the MXene assembly on polymers can’t achieve full coverage. To avoid confusion, we changed the statement to “In contrast, dipping PDMS substrate into pristine MXene solution using the same dipping process resulted in only traces of MXene deposited (Extended Data Fig. 3)” (Lines 110 to 112).

(9) Paragraph starting line 188, could the term depletion be used here?

Response: Exactly. “Deplete” should be better here and we have replaced the “neutralize” with “deplete”.

(10) Is the trend in spacing between the nanosheets consistent with cation radius? If not, why not?

Response: We employed the XRD to calculate the d -spacing between the MXene nanosheets with different salt additions (0.01 mol/L for all cases), as shown below. The results do not present a clear trend with respect to different cation radii. Interlayer spacing of the MXene nanosheets is affected by the solvation of the cations (*Energy & Environmental Science*, 2020, 13, 2549–2558), humidity of the surroundings (*Chemistry of Materials*, 2016, 28(10), 3507-3514; *Journal of Physical Chemistry Letters*, 2015, 6, 4026-4031), ambient temperature and pressure, and the hydrophilicity/hydrophobicity of surface terminations (*Chem. Mater.* 2024, 36(4), 1998-2006).

The MXene used in this study has hydrophilic, negatively charged surfaces and attracts cations with their hydration shells during the self-assembly. During the XRD measurement, the interlayer spacing is further affected by the humidity and temperature in the lab.

Sample	2θ (°)	d-spacing along (002) plane (Å)
Ti ₃ AlC ₂	9.50±0.00	9.30
Ti ₃ C ₂ T _x	7.40±0.06	11.94
Li-Ti ₃ C ₂ T _x	6.25±0.05	14.13
Na-Ti ₃ C ₂ T _x	6.68±0.10	13.22
K-Ti ₃ C ₂ T _x	6.43±0.10	13.73
Cs-Ti ₃ C ₂ T _x	6.45±0.05	13.69
Mg-Ti ₃ C ₂ T _x	6.33±0.08	13.95
Al-Ti ₃ C ₂ T _x	6.05±0.09	14.60

(11) In Fig. 3, are the error bars between different substrate samples or within a single substrate? If the latter, does this indicate roughness?

Response: For Figs. 3c and 3d, the error bars were determined by measuring multiple locations within a single substrate. Those error bars do not necessarily indicate the roughness. For example, the sheet resistance of MXene coating indeed varies for different roughnesses. With increasing the thickness of MXene coating, the sheet resistance usually reaches a static plateau, and the error bar does not change too much. But with increasing thickness, especially in our case, using salt increases the coating roughness, and it fluctuates to some degree because of the possible small-scale aggregation. As PDMS is known for its extremely smooth surface, the decrease in error bars for both coating thickness and sheet resistance with increasing assembly time should not be attributed to a change in roughness but rather to the gradual accumulation of MXene nanosheets on the substrate and, thus the formation of complete coverage leading to a conductive network. This finding is consistent with what has been reported in other dip coating literature (Figure 2b in 10.1002/adfm.202312434, Figure 2c in 10.1016/j.carbon.2020.12.021, and Figure 2a in 10.1016/j.mattod.2020.02.005).

(12) What's the largest scale sample that has been coated? Given that with addition of salt the nanosheets aggregate and needs to be redispersed (as discussed in the experimental), is there a timeframe within which the solution needs to be used? What are the limitations of the processing approach?

Response: The largest sample with MXene coating we have obtained is the MXene-coated Kevlar fabric with an area larger than 300 cm². But for this method, we believe the size of assembly sample will be only limited by the size of the solution container.

The aggregation induced by the salt addition occurs naturally. In our study, after adding the salt, we sonicated the solution for 15 min to redisperse aggregated MXene nanosheets. The redispersed solution can stay stable during the 15 min of assembly. As shown in ED Table 5, the size of MXene nanosheets in the redispersed solution does not change during the assembly process.

We do not see any significant limitations to the SAA method. The aggregation problem caused by adding salt has been resolved by the redispersion process. We have demonstrated that this method is a generic one. It can be applied to a wide range of polymers and salts. It can be scalable to continuous solution-based coating process.

(13) ED Fig 3, can the height profile be changed to give meaningful data to most of the image (the one red spot makes the blue/green non-differentiable)

Response: We shrank the color bar scope to exclude the red spot, as shown below in ED Fig. 3.

Extended Data Fig. 3 | $\text{Ti}_3\text{C}_2\text{T}_x$ nanosheet assembly on PDMS substrates with and without NaCl salt. **a**, Digital images of pure PDMS substrate and $\text{Ti}_3\text{C}_2\text{T}_x$ nanosheet assemblies on PDMS. **b**, Optical image of pure PDMS. **c**, Optical image of $\text{Ti}_3\text{C}_2\text{T}_x$ nanosheet assembly on PDMS without NaCl. Without NaCl salt addition to the $\text{Ti}_3\text{C}_2\text{T}_x$ nanosheet solution, we failed to achieve full coverage and uniform assembly even though there were some nanosheets assembled. **d**, Optical image of $\text{Ti}_3\text{C}_2\text{T}_x$ nanosheet assemblies on PDMS with NaCl. **e**, Optical profilometry map of $\text{Ti}_3\text{C}_2\text{T}_x$ nanosheet assemblies on PDMS with NaCl. Scale bars, 20 μm . The dip-coating assembly time is fixed to be 15 min, the $\text{Ti}_3\text{C}_2\text{T}_x$ nanosheet concentration is 5 mg mL^{-1} , and the NaCl salt concentration is 0.01 mol L^{-1} .

(14) ED Fig. 6 is referenced as showing “uniform coating”, however these are only electron micrographs and don’t show a large area; is there a better technique to complement SEM to show uniform coating?

Response: Yes, we have the large-area optical images (Response letter Fig. 1), which show conformal MXene coatings on different polymers.

Response letter Fig. 1 Optical images of pure polymer substrates and MXene-coated ones.

(15) ED Fig. 7, it would be helpful to show the non-coated fabrics and fibers to highlight the impact of nanosheet assembly. This same comment can be applied to ED Fig. 8 (with just SEM images it's difficult to confirm/support that the nanosheets have been coated)

Response: We have added SEM images of non-coated fabric and fibers, micropillars, and microstrips to the corresponding figures. You can see the clear difference between the coated and non-coated ones in ED Figs. 7 and 8.

Extended Data Fig. 7 | SEM images of Na-Ti₃C₂T_x nanosheets assembled on polymer fibers. **a**, Na-Ti₃C₂T_x nanosheets on different polymer fibers. SEM images show that the Na-Ti₃C₂T_x nanosheets not only wrap the surface of fibers but create bridges to connect the fibers. Also, the assembled Na-Ti₃C₂T_x nanosheets on a single polymer fiber feature a wrinkled structure. Scale bars for top images, 1 mm. Scale bars for bottom images, 20 μm. **b**, non-coated polymer fibers. Scale bars for zoom-out images, 1 mm. Scale bars for insets, 20 μm.

Extended Data Fig. 8 | Assembly of Na-Ti₃C₂T_x nanosheets on micro-patterned and 3D printed PDMS substrates. a, b, SEM images of Na-Ti₃C₂T_x nanosheets assembled on PDMS substrates with micropillar array. c, Non-coated micropillar. d, e, SEM images of Na-Ti₃C₂T_x nanosheets assembled on PDMS substrates with microstrip array. f, Non-coated microstrip. g, h, SEM images of Na-Ti₃C₂T_x nanosheets assembled on 3D printed PDMS substrates. Scale bars for a-d, 10 μm. Scale bars for e-f, 5 μm. Scales bars for g-h, 1 cm.

(16) For the different cations, is there any elemental analysis (such as at% by EDS) that can be used to evaluate composition). Is there a correlation between charge and composition?

Response: Considering the absorption of cations from salt on the MXene surface, we employed vacuum-assisted filtration, as an approximation of the assembled films, to obtain different MXene free-standing films of larger thickness for the SEM-EDS analysis. Also, for each MXene solution, we fixed the salt and MXene concentration to 0.01 mol/L and 10 mg/mL, which is included in 10 mL deionized water. The results of the analyses are tabulated below. We see the presence of cations. However, the accuracy of EDS analysis is within 5 at.% and we cannot make any quantitative conclusions based on the EDS data.

Materials	NaCl-Ti ₃ C ₂ T _x	KCl-Ti ₃ C ₂ T _x	CsCl-Ti ₃ C ₂ T _x	MgCl ₂ -Ti ₃ C ₂ T _x	AlCl ₃ -Ti ₃ C ₂ T _x
Cation amount (at%)	1.1	1.3	1.9	2.3	1.6

(17) From the FTIR spectra of ED Fig. 15, comparison of the -OH stretching frequency intensity is tenuous given changes across different peaks. It would be more reliable if another peak could be used as an internal standard, or this discussion could be softened (or removed)

Response: Thank you for pointing out the problem. We removed ED Fig. 15 and the corresponding discussion to avoid confusion.

(18) ED Fig. 17, state how roughness was determined

Response: The sample was placed under a 50x magnification lens of a Keyence VK-X1000 optical profilometer and evaluated using laser confocal scanning. Plane correction was performed on the scan using the accompanying software before three areas (with slight overlap and together covering the entire scan) were arbitrarily selected for roughness calculations. The areal average roughness (Sa) was chosen as the representative roughness parameter, which is calculated as the mean of the average height difference for the average surface. This description has been updated in lines 576-582 in the Characterization section.

Point-by-Point Response

Reviewer #3

I recommend double check and update the discussion with works focused on layer-by-layer assembly of MXenes, especially to polymer substrates, such as, e.g., <https://onlinelibrary.wiley.com/doi/full/10.1002/smt.202201252> (textile fabric fibers). This is just a suggestion but could increase the understanding of readers in terms of potential impact of the paper.

Response: We agree on adding the reference and we added the reference in the introduction part.